# A Meta-Analysis on the Association between Peptic Ulcer Disease and COVID-19 Severity

**DOI:** 10.3390/vaccines11061087

**Published:** 2023-06-11

**Authors:** Ying Wang, Jie Xu, Liqin Shi, Haiyan Yang, Yadong Wang

**Affiliations:** 1Department of Epidemiology, School of Public Health, Zhengzhou University, Zhengzhou 450001, China; 2Department of Toxicology, Henan Center for Disease Control and Prevention, Zhengzhou 450016, China

**Keywords:** peptic ulcer disease, COVID-19, severity, meta-analysis

## Abstract

The association between peptic ulcer disease and the severity of coronavirus disease 2019 (COVID-19) is inconclusive across individual studies. Thus, this study aimed to investigate whether there was a significant association between peptic ulcer disease and COVID-19 severity through a meta-analysis. The electronic databases (Web of Science, Wiley, Springer, EMBASE, Elsevier, Cochrane Library, Scopus and PubMed) were retrieved for all eligible studies. The Stata 11.2 software was used for all statistical analyses. The pooled odds ratio (OR) with a 95% confidence interval (CI) was calculated by a random-effects meta-analysis model. The heterogeneity was evaluated by the inconsistency index (I^2^) and Cochran’s Q test. Egger’s analysis and Begg’s analysis were conducted to evaluate the publication bias. Meta-regression analysis and subgroup analysis were done to explore the potential source of heterogeneity. Totally, our findings based on confounding variables-adjusted data indicated that there was no significant association between peptic ulcer disease and the higher risk for COVID-19 severity (pooled OR = 1.17, 95% CI: 0.97–1.41) based on 15 eligible studies with 4,533,426 participants. When the subgroup analysis was performed by age (mean or median), there was a significant association between peptic ulcer disease and a higher risk for COVID-19 severity among studies with age ≥ 60 years old (pooled OR = 1.15, 95% CI: 1.01–1.32), but not among studies with age < 60 years old (pooled OR = 1.16, 95% CI: 0.89–1.50). Our meta-analysis showed that there was a significant association between peptic ulcer disease and a higher risk for COVID-19 severity among older patients but not among younger patients.

## 1. Introduction

Coronavirus disease 2019 (COVID-19) was first reported in December 2019 and then rapidly spread to the rest of the world. The World Health Organization (WHO) declared the COVID-19 pandemic on 11 March 2020. The COVID-19 infection was reported to be caused by severe acute respiratory syndrome coronavirus 2 (SARS-CoV-2), which is an enveloped positive-sense single-stranded RNA virus belonging to the coronaviridae family, a subfamily of orthocoronavirinae. As of 1 March 2023, 758,390,564 confirmed cases of COVID-19 infection, including 6,859,093 deaths, have been reported to the WHO. It is reported that the most common clinical signs are heterogeneous and can range from entirely asymptomatic illness to mild or severe flu-like symptoms such as fever, cough, headache, sore throat, myalgia, dyspnea, vomiting, diarrhea, nausea and loss of smell and taste, to acute respiratory distress syndrome and even death [1]. Although the mortality of COVID-19 infection was reported to be 5% on average, the risk of severe/critical infection and death rises with advanced age and in the presence of several comorbidities such as hypertension, diabetes mellitus, cerebrovascular disease, cardiovascular disease, chronic respiratory illness, chronic liver disease, chronic renal disease, cancer, dementia, autoimmune disease, and so on [2,3,4,5,6,7,8,9].

Although COVID-19 was initially recognized as a respiratory illness, SARS-CoV-2 can damage many organ systems. Approximately 20% of infected patients develop gastrointestinal symptoms, particularly abdominal pain, bloody diarrhea or non-bloody diarrhea [10]. The virus is detected in the stool samples of more than one-third of patients with pulmonary symptoms, and it may persist for weeks to months after the onset of COVID-19-related symptoms [11]. Meanwhile, the detection of SARS-CoV-2 RNA and intracellular staining for viral nucleocapsid protein in gastric and duodenal epithelium indicated infection of these gastrointestinal gland epithelial cells by SARS-CoV-2 [12]. The spike protein of SARS-CoV-2 consists of two subunits, S1 and S2, which are essential for viral entry into cells. The S1 subunit binds the angiotensin-converting enzyme-2 (ACE2) receptor, whereas the S2 subunit is cleaved by transmembrane serine protease-2 (TMPRSS2), thereby facilitating viral fusion with the cell membrane [13]. Studies have shown that both ACE2 and TMPRSS2 are highly expressed in the gastrointestinal tract, supporting the entry of SARS-CoV-2 into the host cells and explaining why this system is commonly affected [14]. In addition to being a binding receptor for SARS-CoV-2, ACE2 is a key regulator of the renin-angiotensin system (RAS), which controls blood pressure and inflammation [15]. It is worth noting that COVID-19 causes RAS dysregulation and affects the host inflammatory response [15].

Peptic ulcer disease affects four million people worldwide annually and has an estimated lifetime prevalence of 5–10% in the general population [16]. Peptic ulcer disease mainly contains gastric ulcers and duodenal ulcers. Studies have revealed that the odds of mortality in emergently admitted geriatric patients with acute gastric ulcers were two times higher than in adult patients. The following comorbidities (such as alcohol abuse, deficiency anemias, chronic blood loss, chronic heart failure, chronic pulmonary disease, hypertension, fluid/electrolyte disorders, uncomplicated diabetes mellitus and renal failure) were reported to be associated with gastric ulcers [17]. A perforated peptic ulcer is a severe complication of peptic ulcer disease, and individuals with perforated peptic ulcer also have severe abdomen pain with a high risk of long-term illness and fatality [18].

A large amount of evidence has indicated that vaccinations are particularly effective in reducing the incidence of severity and mortality among COVID-19 patients [19,20,21,22,23,24,25,26,27,28]. To date, more than 200 COVID-19 vaccine candidates are in various stages of development, and more than 50 of these candidates have started clinical trials. These candidates may use inactivated or live attenuated viruses, nucleic acids (DNA or RNA), viral vectors, recombinant proteins or virus particles [29,30,31,32,33,34,35,36,37]. WHO is committed to keeping the momentum for boosting access to COVID-19 vaccinations going and will continue to assist countries in expediting vaccine delivery in order to save lives and prevent people from becoming seriously ill. In addition, identifying risk factors is crucial for prioritizing people who should get vaccinations. Sex, age, and certain past medical conditions [38,39,40,41,42,43] have all been discovered to be risk factors for the severity and mortality of COVID-19 patients up until this point. It is not clear whether peptic ulcer disease was a risk factor for COVID-19 severity.

A series of published studies have investigated the association between peptic ulcer disease and the risk of disease severity in patients with COVID-19 infection; however, the conclusions from previous individual studies are still contradictory. A recent large-sample-sized retrospective study by Meister et al. [44] found that patients with COVID-19 infection and peptic ulcer disease had a significantly higher risk for severity in comparison to COVID-19 patients without peptic ulcer disease. In line with the study by Meister et al., St Sauver et al. [45] also observed that COVID-19 patients with peptic ulcer disease had a significantly higher risk of mortality in comparison to those without peptic ulcer disease. But some other studies reported the opposite findings: COVID-19 patients with peptic ulcer disease did not have a significantly higher risk for severity in comparison to those without peptic ulcer disease [46,47]. To our knowledge, no comprehensive meta-analysis has been performed on this pivotal topic so far; we, therefore, performed this meta-analysis to determine whether there was a significant association between peptic ulcer disease and COVID-19 severity. Considering age and comorbidities are recognized as known confounding factors for disease progression in COVID-19 patients [2,3,4,5,6,7,8,9], which might modify the relation between peptic ulcer disease and COVID-19 severity, the pooled effects were synthesized using the confounding variables-adjusted data in this current meta-analysis.

## 2. Materials and Methods

We performed this meta-analysis in light of the preferred reporting item for systematic reviews and meta-analysis (PRISMA) statement documented by Liberati et al. [48]. The online databases (Web of Science, Wiley Library, Springer Link, EMBASE, Elsevier ScienceDirect, Cochrane Library, Scopus and PubMed) were retrieved for all eligible studies as of 30 January 2023, using the keywords: (“peptic ulcer” or “digestive ulcer” or “gastric ulcer” or “duodenal ulcer” or “gastrointestinal ulcer”) and (“2019-nCoV” or “COVID-19” or “SARS-CoV-2” or “2019 novel coronavirus” or “coronavirus disease 2019” or “severe acute respiratory syndrome coronavirus 2”). The outcome of interest was defined as severity (including severe/critical disease, the requirement for invasive mechanical ventilation, severity/progression, admission to the intensive care unit and death). The reference lists of the previously published review papers and the retrieved original literature were also thoroughly examined to identify all potentially relevant articles as completely as possible [49,50,51]. Two authors (Ying Wang and Jie Xu) independently performed a literature search. Any discrepancy was resolved by consulting a third author. The control group consisted of COVID-19 patients who did not have peptic ulcer disease, and the exposure group consisted of COVID-19 patients who had peptic ulcer disease.

All articles were deemed eligible for inclusion if they: (i) investigated the association between peptic ulcer disease and COVID-19 severity based on confounding variables-adjusted data; (ii) were published in English; (iii) were peer-reviewed and (iv) reported confirmed COVID-19 patients. The articles were excluded if they: (i) were review papers, animal studies, study protocol, comments, editorials, preprints, errata, duplicate literature or case reports, and (ii) reported confounding variables-unadjusted data on the association of peptic ulcer disease with COVID-19 severity.

The main information extracted from each eligible study included author name, country, study design, sample size, mean or median age, male proportion, adjusted risk factors, adjusted effect estimates and their corresponding 95% confidence interval (CI). Two authors (Ying Wang and Liqin Shi) independently performed data extraction. Any discrepancy was resolved by consulting a third author. The quality of the included studies was evaluated by investigators according to the Newcastle-Ottawa Scale (NOS) (Appendix A). High-quality studies are those with scores of 8 or more, medium-quality studies with scores of 5 to 7, and low-quality studies with scores of 4 or below [52].

The Stata 11.2 software (StataCorp, College Station, TX, USA) was used to conduct all statistical analyses. The heterogeneity across the included studies was evaluated by the inconsistency index (I^2^) and Cochran’s Q test. The pooled effect size (which was presented as an odds ratio (OR)) with 95% CI was calculated by using a random-effects statistical model. Both Egger’s linear regression analysis and Begg’s rank correlation analysis were conducted to evaluate the possibility of publication bias. Meta-regression analysis and subgroup analysis by sample size, male proportion, study design, age, quality rating, and setting were done to explore the potential source of heterogeneity. A (two-sided) *p*-value of <0.05 was regarded as statistical significance.

## 3. Results

Up to 589 articles were searched in the electronic databases, and 10 additional articles were identified through the lists of references. Further, 390 duplicate articles were excluded; 156 articles were excluded after reading the titles and abstracts, and 38 articles were excluded after scanning the full texts. At last, a total of 15 eligible studies [44,45,46,47,53,54,55,56,57,58,59,60,61,62,63] consisting of 4,533,426 patients with COVID-19 were included in this meta-analysis according to the inclusion and exclusion criteria aforementioned. The flowchart of the study selection is illustrated in Figure 1. There were four studies conducted in the United States of America, four studies conducted in South Korea, two studies conducted in the United Kingdom, two studies conducted in Austria and one each in Spain, Croatia and Estonia. In terms of study design, there were thirteen retrospective studies and two cohort studies. The sample sizes across the included studies ranged from 1544 to 3,604,812 cases. The male proportion among the individual studies varied from 26.77% to 56.2%. The age (mean/median) among the eligible studies varied from 35.8 to 86.7 years old. The main characteristics of the included studies are demonstrated in Table 1.

Totally, our findings based on confounding variables-adjusted data indicated that there was no significant association between peptic ulcer disease and the higher risk for COVID-19 severity (pooled OR = 1.17, 95% CI: 0.97–1.41, Figure 2A). We did not observe a significant association between peptic ulcer disease and the higher risk for COVID-19 severity in the subgroup analyses stratified by male proportion (pooled OR = 1.25, 95% CI: 0.99–1.58 for male proportion ≥ 50% and 1.13, 95% CI: 0.87–1.45 for male proportion < 50%), study design (pooled OR = 1.19, 95% CI: 0.98–1.44 for retrospective studies and 1.03, 95% CI: 0.72–1.47 for the others) and setting (pooled OR = 1.10, 95% CI: 0.98–1.22 for hospitalized COVID-19 patients and 1.22, 95% CI: 0.93–1.58 for all COVID-19 patients). When the subgroup analysis was performed by age (mean or median), there was a significant association between peptic ulcer disease and the higher risk for COVID-19 severity among studies with age ≥60 years old (pooled OR = 1.15, 95% CI: 1.01–1.32) (Figure 2B), but not among studies with age < 60 years old (pooled OR = 1.16, 95% CI: 0.89–1.50) (Figure 2C). Meta-regression exhibited that sample size (*p* = 0.111), male proportion (*p* = 0.691), age (*p* = 0.877), study design (*p* = 0.651), quality rating (*p* = 0.224), and setting (*p* = 0.676) might not be the potential sources of heterogeneity. Both Begg’s test (*p* = 0.428, Figure 3A) and Egger’s test (*p* = 0.751, Figure 3B) revealed that no potential publication bias existed in the study.

## 4. Discussion

Peptic ulcer disease is a frequent and dangerous disorder that can result in life-threatening consequences such as major bleeding or bowel rupture [64], and thus it is assumed to increase the risk of severity of COVID-19 patients. In this current meta-analysis based on confounding variables-adjusted data, we did not observe a significant association between peptic ulcer disease and COVID-19 severity. Further subgroup analysis stratified by mean/median age revealed that peptic ulcer disease was statistically significantly related to COVID-19 severity among older patients but not among younger patients. Age has been reported to be the most important risk factor for COVID-19 severity and mortality [44,65]. A previous study by Yang et al. demonstrated that chronic liver disease was significantly independently associated with an increased risk for severity and mortality among elderly individuals with COVID-19 infection [66]. Li et al. observed that a significant association between myocardial infarction and the increased risk for COVID-19 mortality did exist among studies with patients who are ≥60 years old [2]. Ren et al. observed that peripheral artery disease significantly increased the risk for mortality among COVID-19 patients in the subgroup of patients with a mean age of ≥60 years old [67]. Oguz et al. documented that age-related immune changes might be at the bottom of the severe course of COVID-19, and age-related hormonal changes might have considerable importance due to their interactions with these immune modifications, as well as endothelial dysfunction and concomitant cardiometabolic illnesses [68]. The exact reason for the association between certain underlying diseases and COVID-19 severity among older individuals was not clear, which should be explored in the future. Similarly, Batsiou, A. et al. reviewed SARS-CoV-2 infection and outcomes in children with inflammatory bowel disease (IBD) and showed that IBD patients with SARS-CoV-2 infection did not have a more severe clinical course or disease progression [69].

In addition to patients with peptic ulcer disease, which is thought to increase the risk of severe disease in patients with COVID-19, SARS-CoV-2 infection can also cause several gastrointestinal symptoms, including diarrhea, nausea, vomiting, and abdominal discomfort [70]. A meta-analysis based on adjusted effect estimates demonstrated that gastrointestinal symptoms were not significantly associated with the risk of mortality for COVID-19 patients [71]. Current studies have shown a connection between COVID-19 infection and peptic ulcer disease. A study by Merdad et al. reported a patient who presented to the hospital with classic complaints suggestive of complicated peptic ulcer disease without any respiratory symptoms [70]. Melazzini et al.’s study showed that five patients with COVID-19 were diagnosed with peptic ulcer disease on admission without any previous history of peptic ulcer [72]. He et al. reported the case of a COVID-19 patient who developed a duodenal bulb ulcer after weeks of infection [73]. Deb et al. reported three cases of patients with COVID-19 infection who developed massive gastrointestinal bleeding from gastric ulcers during their hospitalization despite being on proton-pump inhibitor prophylaxis [74]. In addition, some other studies investigated the impact of the COVID-19 pandemic on the incidence of peptic ulcer disease. For example, research by Dao et al. observed that the proportion of peptic ulcer disease was significantly higher during the time of the state of emergency due to the ongoing COVID-19 pandemic in 2020 when compared to 2019 at the same health facility in Vietnam [75]. In line with Dao et al.’s study, Jian et al. observed that the rate of peptic ulcer disease increased dramatically in two hospitals in Wuhan, China, in 2020. They also observed an increase in the incidence of severe peptic ulcer disease after the pandemic in comparison to the same period before the pandemic [76]. The negative impact of COVID-19 on peptic ulcer disease should be focused on in the future as well.

To the best of our knowledge, this is the first study to quantitatively evaluate the association between peptic ulcer disease and the risk for COVID-19 severity by using a meta-analysis. However, some limitations should be noted in this study. Firstly, the pooled effects were synthesized by using the confounding variables-adjusted data (chiefly controlling sex, age, hypertension, diabetes mellitus, cardiovascular disease, cerebrovascular disease, renal disease, liver disease, chronic pulmonary disease, etc.), but additional confounding variables (such as types and severity of peptic ulcer disease, medications, vaccination status and SARS-CoV-2 variants) [77,78,79,80,81,82,83,84] might certainly have significant impacts on the relationship of peptic ulcer disease with COVID-19 severity. To our knowledge, none of the included studies offered details on the types and severity of peptic ulcer disease, medications, vaccination status or SARS-CoV-2 variants. This hindered us from performing further analysis to evaluate the influences of medications, types and severity of peptic ulcer disease, vaccination status and SARS-CoV-2 variants on the relationship between peptic ulcer disease and COVID-19 severity. Secondly, there was statistical heterogeneity across the included studies; thus, we performed meta-regression analysis to identify potential sources of heterogeneity, but the tested variables might not contribute to the heterogeneity. Thirdly, the included studies were mainly from South Korea and the United States of America; thus, the results were interpreted with caution in other countries or regions. Fourthly, most of the included studies were designed retrospectively; future prospective studies with large sample sizes are warranted to verify our findings.

## 5. Conclusions

Our findings based on confounding variables-adjusted data showed that there was a significant association between peptic ulcer disease and a higher risk for COVID-19 severity among older patients but not among younger patients. Further, well-designed studies with more data are required to verify our findings.

## Figures and Tables

**Figure 1 vaccines-11-01087-f001:**
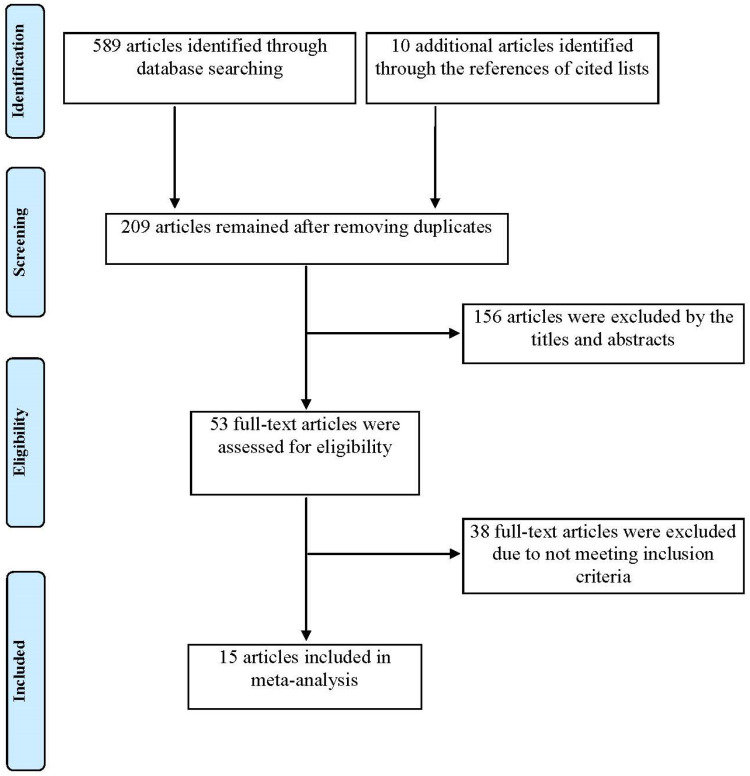
The flowchart of study selection in this meta-analysis.

**Figure 2 vaccines-11-01087-f002:**
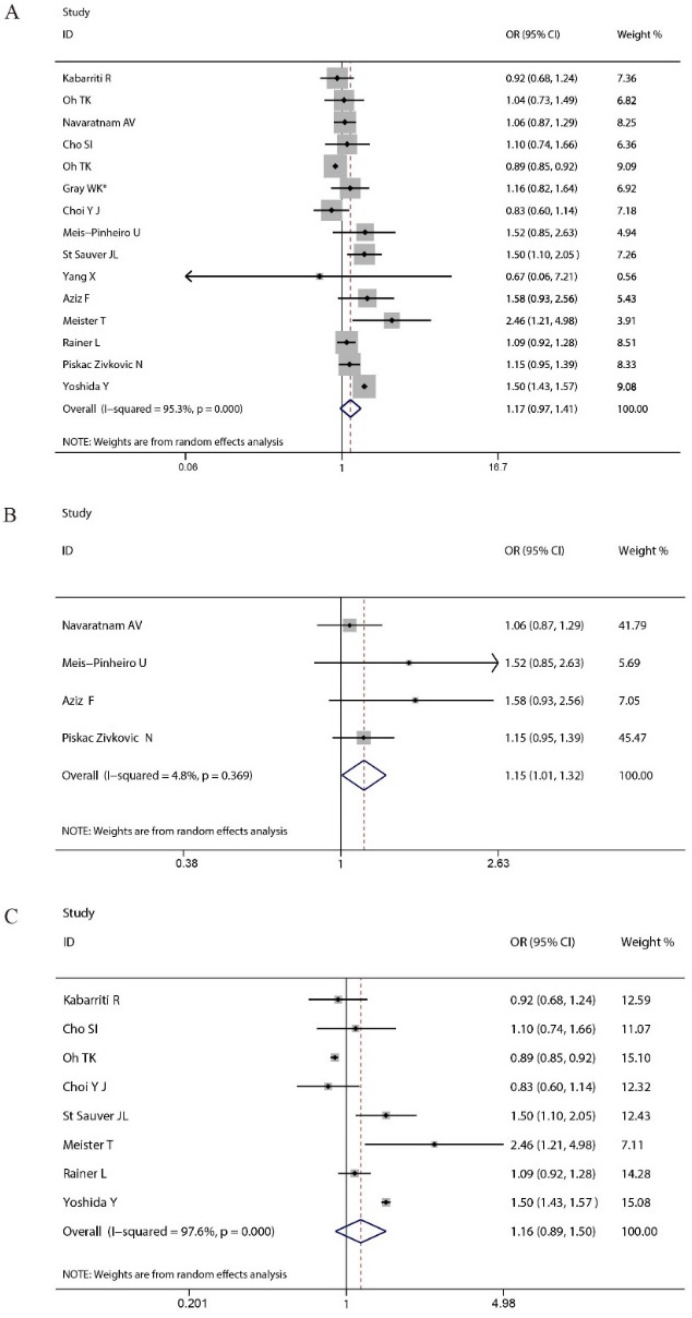
The forest plots demonstrated that there was no significant association between peptic ulcer disease and a higher risk for COVID-19 severity (**A**); Subgroup analysis by age showed that there was a significant association between peptic ulcer disease and a higher risk for COVID-19 severity among studies with age ≥60 years old (**B**), but not among studies with age <60 years old (**C**). * indicates that the combined value was calculated based on data from subgroups.

**Figure 3 vaccines-11-01087-f003:**
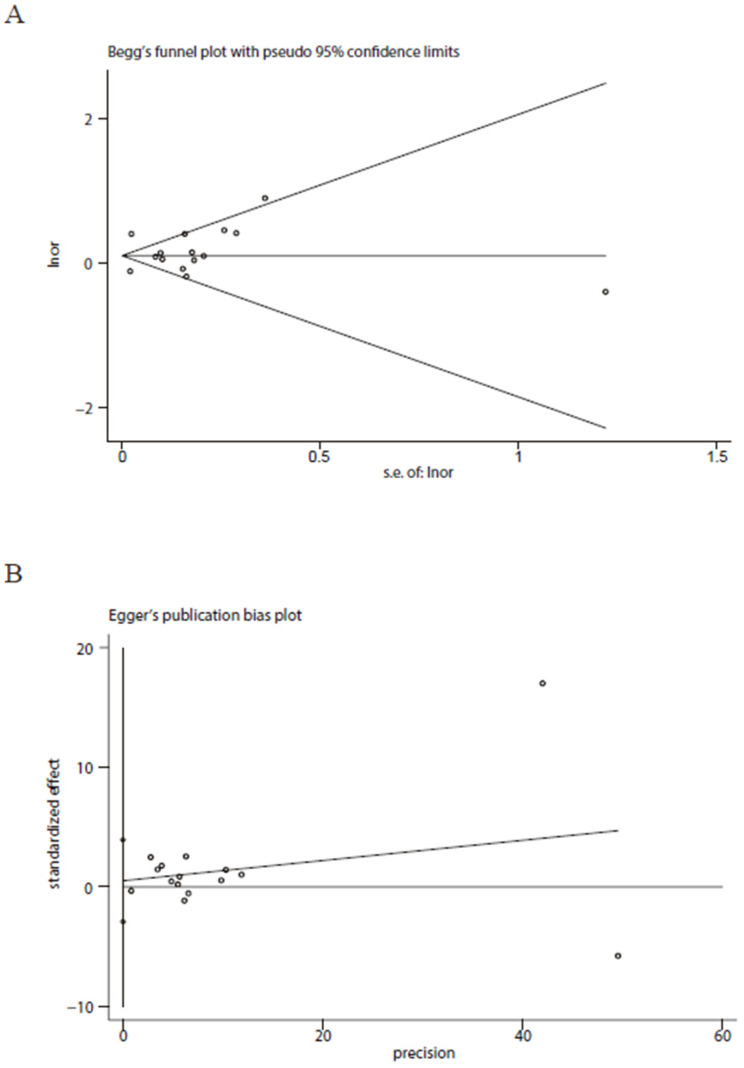
Begg’s test (**A**) and Egger’s test (**B**) showed that there was no potential publication bias in this meta-analysis.

**Table 1 vaccines-11-01087-t001:** Characteristics of the included studies in this meta-analysis.

Author	Country	Data Collection Time (Year/Month/Day)	Sample Size	Male (%)	Age (Years)	Study Design	Adjusted-Effect (95% CI)	Setting	Adjusted Risk Factors	Score #
Kabarriti R [54]	USA	2020/03/14~2020/04/27	5902	46.9	57.5	Retrospective cohort study	0.92 (0.68–1.24)	Hospitalized COVID-19 patients	Age, sex, socioeconomic status, ethnicity/race, body mass index, hypertension, cardiovascular disease, diabetes mellitus, cancer, liver disease, dementia, chronic pulmonary disease, hemiplegia or paraplegia, kidney disease and human immunodeficiency virus/acquired immune deficiency syndrome	5
Oh TK [55]	Korea	2020/01/01~2020/06/26	7780	NA	NA	Cohort study	1.04 (0.73–1.49)	Hospitalized COVID-19 patients	Hypertension, diabetes mellitus without chronic complication, diabetes mellitus with chronic complication, peripheral vascular disease, renal disease, rheumatic disease, dementia, hemiplegia or paraplegia, moderate or severe liver disease, mild liver disease, cerebrovascular disease, congestive heart failure, myocardial infarction, malignancy, metastatic solid tumor and human immunodeficiency virus/acquired immune deficiency syndrome	8
Navaratnam AV [53]	UK	2020/03/01~2020/05/31	91,541	55.4	72.58	Retrospective cohort study	1.056 (0.865–1.289)	Hospitalized COVID-19 patients	Sex, deprivation score, ethnicity, date of discharge and Charlson comorbidity index items (peripheral vascular disease, congestive heart failure, acute myocardial infarction, cerebrovascular disease, dementia, chronic pulmonary disease, connective tissue disease or rheumatic disease, mild liver disease, moderate or severe liver disease, diabetes mellitus without chronic complications, diabetes mellitus with chronic complications, paraplegia and hemiplegia, renal disease, primary cancer, metastatic carcinoma and obesity)	5
Cho SI [47]	Korea	2020/05/15	7590	40.8	46.0 ± 19.6	Retrospective cohort study	1.10 (0.74–1.66)	All COVID-19 patients	Age and sex	6
Oh TK [63]	Korea	2020/01/01~2020/08/27	7713	39.5	48.4	Retrospective cohort study	0.89 (0.85–0.92)	All COVID-19 patients	Age, sex, annual income level in 2020, residence in 2010, underlying disability, Charlson comorbidity index, hypertension, myocardial infarction, congestive heart failure, peripheral vascular disease, cerebrovascular disease, diabetes mellitus without chronic complications, diabetes mellitus with chronic complications, renal disease, hemiplegia or paraplegia, rheumatic disease, mild liver disease, moderate to severe liver disease, chronic pulmonary disease, any cancer, metastatic solid tumor and human immunodeficiency virus/acquired immune deficiency syndrome	8
Gray WK * [57]	UK	2020/03/01~2020/09/30	101,632	NA	NA	Retrospective cohort study	1.16 (0.82–1.64)	Hospitalized COVID-19 patients	Age band, sex, deprivation quintile, ethnicity and Charlson comorbidity index items (congestive heart failure, peripheral vascular disease, acute myocardial infarction, cerebrovascular disease, dementia, chronic pulmonary disease, connective tissue disease/rheumatic disease, mild liver disease, moderate or severe liver disease, diabetes mellitus without chronic complications, diabetes mellitus with chronic complications, paraplegia and hemiplegia, renal disease, obesity, primary cancer and metastatic carcinoma)	5
Choi YJ [58]	Korea	2020/05/15	7590	40.8	46.61	Retrospective cohort study	0.831 (0.604–1.143)	All COVID-19 patients	Age, sex and prevalence of underlying disease (myocardial infarction, congestive heart failure, peripheral vascular disease, cerebrovascular disease, dementia, chronic pulmonary disease, rheumatologic disease, mild liver disease, diabetes mellitus without chronic complications, diabetes mellitus with chronic complications, paralysis (hemiplegia or paraplegia), renal disease, malignancy, moderate or severe liver disease, metastatic solid tumor and acquired immune deficiency syndrome)	4
Meis-Pinheiro U [46]	Spain	2020/03/01~2020/05/31	2092	26.77	86.7 ± 7.06	Retrospective cohort study	1.52 (0.85–2.63)	All COVID-19 patients	Age, pneumonia, fever, dyspnea, stupor, refusal to oral intake, diarrhea, mucous secretion, dry cough, eczema, asthenia, muscle pains, nasal congestion, sore throat, vomiting, lip blisters, confusion, insomnia, dementia, hepatopathy, cardiovascular disease, cerebrovascular disease, diabetes mellitus without organic involvement, diabetes mellitus with organic involvement, chronic obstructive pulmonary disease, chronic kidney disease, connective tissue disease, cancer without metastases, cancer with metastases and hematological tumor	4
St Sauver JL [45]	USA	2020/03/01~2020/09/30	9928	47.9	35.8	Retrospective cohort study	1.50 (1.10–2.05)	All COVID-19 patients	Age (continuous variable), sex, race, ethnicity, body mass index category and smoking status	4
Yang X [59]	USA	2020/01/01~2021/05/08	1544	NA	NA	Cohort study	0.67 (0.06–7.21)	All COVID-19 patients	Social demographics (age, sex, race, and ethnicity), lifestyle factors (body mass index, and smoking status), comorbidities (hemiplegia or paraplegia, dementia, liver disease, myocardial infarction, congestive heart failure, chronic pulmonary disease, cancer, diabetes mellitus, stroke, peripheral vascular disease, rheumatologic disease, and renal disease) and month of COVID-19 diagnosis	8
Aziz F [60]	Austria	2020/03~2021/03	40,602	52.4	72.95	Retrospective cohort study	1.58 (0.93–2.56)	Hospitalized COVID-19 patients	Sex, age, intensive care unit admission, myocardial infarction, cardiac arrhythmias, valvular heart disease, hypertension, congestive heart failure, peripheral vascular disease, stroke, chronic obstructive pulmonary disease, dementia, liver disease, other neurological disorders, renal disease, hypothyroidism, fluid and electrolyte disorders, deficiency anemia, depression, and Charlson comorbidity index	5
Meister T [44]	Estonia	2020/02/26~2021/02/28	66,295	55.99	44.1 ± 20.6	Retrospective cohort study	2.46 (1.21–4.98)	All COVID-19 patients	Sociodemographic characteristics (gender, and age), pre-COVID-19 comorbidity (Charlson index score, acute myocardial infarction, congestive heart failure, peripheral vascular disease, cerebrovascular disease, dementia, chronic pulmonary disease, rheumatologic disease, chronic kidney disease, acquired immune deficiency syndrome, diabetes mellitus, cancer, liver disease, obesity, sleep apnea, hyperlipidemia, hypertension) and influenza and vaccination within 2 years before COVID-19 (Flu vaccine during last 2 years and influenza during last 2 years)	6
Rainer L [56]	Austria	2020/02~2021/12	3,604,812	45.5	41.75	Retrospective cohort study	1.09 (0.92–1.28)	All COVID-19 patients	Age group, sex, and healthcare region	7
Piskac Zivkovic N [61]	Croatia	2020/03~2021/03	4014	56.20	74 (64–82)	Retrospective cohort study	1.15 (0.95–1.39)	Hospitalized COVID-19 patients	Age and sex	4
Yoshida Y [62]	USA	2020/01/01~2021/12/31	574,391	46.6	52.3 ± 18.5	Retrospective cohort study	1.50 (1.43–1.57)	All COVID-19 patients	Age, race, ethnicity, visit type, and any medication use	5

Note: The age (year) was presented as mean ± standard deviation (SD) or median (interquartile range); CI: confidence interval; NA: not available; COVID-19: coronavirus disease 2019; UK: the United Kingdom; USA: the United States of America; * indicates that the combined value was calculated based on data from subgroups; # scores for quality assessment of the included studies according to the Newcastle–Ottawa scale.

## Data Availability

The data that support the findings of this study are included in this article and are available from the corresponding authors upon reasonable request.

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
