# Peer review of "A Meta-Analysis on the Association between Peptic Ulcer Disease and COVID-19 Severity"

_vaccines, 2023, doi:10.3390/vaccines11061087_

Round 1

Reviewer 1 Report

This meta-analysis based on 15 included studies with 4,533,426 participants showed that there was a significant association between peptic ulcer disease and a higher risk for COVID-19 severity among older patients, but not among younger patients. The results of this study are valuable to the readers, while the  statistical analysis was  carried out properly. Specifically, authors evaluated the heterogeneity of the included studies using the consistency index I2 and the Cochran’s Q test, while Egger’s analysis and Begg’s analysis were also  conducted to evaluate any publication bias. Finally, meta-regression analysis and subgroup analysis were done to explore the potential source of heterogeneity. I believe this study should be published since it has a clear message. The number of participants is large enough, while the authors tried to minimize any bias from potential confounding factors. The only recommendation, is that I believe that it would be beneficial for the authors to to discuss the following recent systematic review:

Batsiou, A.; Mantzios, P.; Piovani, D.; Tsantes, A.G.; Kopanou Taliaka, P.; Liakou, P.; Iacovidou, N.; Tsantes, A.E.; Bonovas, S.; Sokou, R. SARS-CoV-2 Infection and Outcomes in Children with Inflammatory Bowel Diseases: A Systematic Review. J. Clin. Med. 202211, 7238. https://doi.org/10.3390/jcm11237238

Reviewer 2 Report

- Interesting meta analysis on prevalence of Peptic Ulcer disease and Covid-19.

There are a few changes I would like to suggest:

- Line 30: The last word should be declared instead of declares.

- Line 45: Since the authors don't indicate which year they are taking about using a present tense rather than past would be grammatically correct. I would paraphrase it saying: Peptic ulcer disease affects four million people worldwide annually and has an estimated lifetime prevalence of 5-10% in the general population (10).

Analytical suggestions:

- The studies have not been assessed for influence diagnosis. Though publication bias has been assessed with Begg's and Egger's tables, regression test can be added

- Table 1 is very informative and summarizes a lot of detailed information.

Reviewer 3 Report

Dear Authors,

Although this meta-analysis adds significant information to the field, I would like to see a revised version of the manuscript according to the following recommendations.

In general, the manuscript would benefit from careful editing throughout to ensure full sentences and proper grammar are used. A native English speaker could improve the quality of the manuscript.

Introduction

Introduction should include details regarding the mechanism that peptic ulcer affects COVID-19 severity.

Since the aim of the journal is the role of Vaccines, I would like to see a paragraph about the way that your meta-analysis is connected with COVID-19 vaccines.

Methods

Wiley, Springer, and Elsevier are not electronic databases but publishing companies. Authors should update their literature search by including more databases such as Scopus.

Regarding the search algorithm how did you the Boolean Operators to connect the keywords?

In the sentence “The reference lists of the previously published review papers and the retrieved original literature were also thoroughly examined to identify all potentially relevant articles as completely as possible.” please add references.

Please, add risk of bias assessment and consider then quality of studies as a potential source of heterogeneity performing meta-regression or subgroup analysis.

Please, add data collection time in Table 1.

Please, clarify whether retrospective studies are cohort or case-control.

Please, use sample size in meta-regression analysis.

Please, add the reasons that you exclude papers in Figure 1.

Discussion

In general, you should re-write the Discussion section.

In particular, the first paragraph in the Discussion section is about vaccination against COVID-19. Could you please explain the rationale behind this paragraph? I cannot understand what this paragraph offers to the manuscript. Please, clarify the way that your meta-analysis is connected with COVID-19 vaccination.

The fourth paragraph in the Discussion section is mainly about case reports. Since you performed a meta-analysis you should not discuss your findings with case reports of low quality. Moreover, Limitations should be added last.

What is the public health impact of your findings? What healthcare providers and patients could take as a message from your meta-analysis?

Please, specify the way that future studies could overcome the existing bias.

Author Response

Dear Authors,Although this meta-analysis adds significant information to the field, I would like to see a revised version of the manuscript according to the following recommendations.

Point 1: In general, the manuscript would benefit from careful editing throughout to ensure full sentences and proper grammar are used. A native English speaker could improve the quality of the manuscript.

Response 1: Thank you very much for your constructive suggestions and comments. We have invited native English speakers to revise our articles to improve the quality of our manuscript. Thanks again. 

IntroductionPoint 2: Introduction should include details regarding the mechanism that peptic ulcer affects COVID-19 severity.

Response 2: Thank you very much for your constructive suggestions and comments. In the revised manuscript, we have added possible mechanisms of interaction between the gastrointestinal system and COVID-19. “Although COVID-19 was initially recognized as a respiratory illness, SARS-CoV-2 has capability to damage many organ systems. Approximately 20% of infected patients develop gastrointestinal symptoms, particularly abdominal pain, bloody diarrhea, or non-bloody diarrhea. Virus is detected in the stool samples of more than one third of patients with pulmonary symptoms, and these viruses may persist for weeks to months after the onset of COVID-19 related symptoms. Meanwhile, the detection of SARS-CoV-2 RNA and intracellular staining for viral nucleocapsid protein in gastric and duodenal epithelium indicated infection of these gastrointestinal gland epithelial cells by SARS-CoV-2. The spike protein of SARS-CoV-2 consists of two subunits, S1 and S2, that are essential for viral entry into cells. The S1 subunit binds the angiotensin converting enzyme-2 (ACE2) receptor, whereas S2 is cleaved by transmembrane serine protease-2 (TMPRSS2), thereby facilitating viral fusion with the cell membrane. Studies have shown that both ACE2 and TMPRSS2 are highly expressed in the gastrointestinal tract, supporting the entry of SARS-CoV-2 into the host cells and explaining why this system is commonly affected. In addition to being a binding receptor for SARS-CoV-2, ACE2 is a key regulator of the renin-angiotensin system (RAS), which controls blood pressure and inflammation. It is worth noting that COVID-19 causes RAS dysregulation and affects the host inflammatory response.”

 Point 3: Since the aim of the journal is the role of Vaccines, I would like to see a paragraph about the way that your meta-analysis is connected with COVID-19 vaccines.

Response 3: Thank you very much for your constructive suggestions and comments. In the revised manuscript, we have added a paragraph about the relationship between our meta-analysis and vaccines. “A large amount of evidence has indicated that vaccinations have shown to be particularly effective in reducing the incidence of severity and mortality among COVID-19 patients. To date, more than 200 COVID-19 vaccine candidates are in various stages of development, and more than 50 of these candidates have started clinical trials. These candidates may use inactivated or live attenuated viruses, nucleic acids (DNA or RNA), viral vectors, and recombinant proteins or virus particles. WHO is committed to keep the momentum for boosting access to COVID-19 vaccinations going, and will continue to assist countries in expediting vaccine delivery in order to save lives and save people from becoming seriously ill. In addition, identifying risk factors is crucial for prioritizing people who should get vaccinations. Sex, age, and certain past medical conditions have all been discovered to be risk factors for the severity and mortality of COVID-19 patients up until this point. It is not clear whether peptic ulcer disease was a risk factor for COVID-19 severity.”. 

MethodsPoint 4: Wiley, Springer, and Elsevier are not electronic databases but publishing companies. Authors should update their literature search by including more databases such as Scopus.

Response 4: Thank you very much for your constructive suggestions and comments. Although Wiley, Springer, and Elsevier are not electronic databases but publishing companies, these publishing companies have published a substantial amount of COVID-19 related research. In order to make our included literature more complete, we also searched the literature published by these major publishing companies. We appreciate your suggestion to broaden the literature search by including additional databases such as Scopus. In the revised manuscript, we have added the Scopus database and re-conducted the literature search. As of January 30, 2023, no missed studies were found according to our inclusion and exclusion criteria. Thanks again. 

Point 5: Regarding the search algorithm how did you the Boolean Operators to connect the keywords?

Response 5: Thank you very much for your constructive suggestions and comments. Our search strategy was as follows: (“peptic ulcer” OR “digestive ulcer” OR “gastric ulcer” OR “duodenal ulcer” OR “gastrointestinal ulcer”) AND (“2019-nCoV” OR “COVID-19” OR “SARS-CoV-2” OR “2019 novel coronavirus” OR “coronavirus disease 2019” OR “severe acute respiratory syndrome coronavirus 2”). We have added these in the revised manuscript. Thanks again. 

Point 6: In the sentence “The reference lists of the previously published review papers and the retrieved original literature were also thoroughly examined to identify all potentially relevant articles as completely as possible.” please add references.

Response 6: Thank you very much for your constructive suggestions and comments. In the revised manuscript, we have added references. Thanks again. 

Point 7: Please, add risk of bias assessment and consider then quality of studies as a potential source of heterogeneity performing meta-regression or subgroup analysis.

Response 7: Thank you very much for your constructive suggestions and comments. The studies we included were cohort studies, so we assessed the quality of each included study by the Newcastle-Ottawa-Scale (NOS), which ranges from zero to nine stars, as detailed in Supplementary Table 1. We have analyzed meta-regressions to explore possible sources of heterogeneity by quality rating, and our result show that quality rating (P = 0.224) might not be the potential sources of heterogeneity. Thanks again.

 Point 8: Please, add data collection time in Table 1.

Response 8: Thank you very much for your constructive suggestions and comments. In the revised manuscript, we have added data collection time to each of the included studies in Table 1. Thank you again for your comments. 

Point 9: Please, clarify whether retrospective studies are cohort or case-control.

Response 9: Thank you very much for your constructive suggestions and comments. We are very sorry that we did not clearly describe the study design of the included studies. We carefully reviewed each included study and determined that all included retrospective studies were cohort studies. In the revised manuscript, we have revised the contents in Table 1. 

Point 10: Please, use sample size in meta-regression analysis.

Response 10: Thank you very much for your constructive suggestions and comments. We have analyzed meta-regressions to explore possible sources of heterogeneity by sample size, and our result show that sample size (P = 0.111) might not be the potential sources of heterogeneity. Thank you again for your comments. 

Point 11: Please, add the reasons that you exclude papers in Figure 1.

Response 11: Thank you very much for your constructive suggestions and comments. The criteria for inclusion and exclusion of articles in Figure 1 are as follows: all articles were deemed eligible for inclusion if they (i) investigated the association between peptic ulcer disease and COVID-19 severity on the basis of confounding variables-adjusted data, (ii) were published in English, (iii) were peer-reviewed, and (iv) reported confirmed COVID-19 patients. Articles were excluded if they (i) were review papers, animal studies, study protocol, comments, editorials, preprints, errata, duplicate literature, case reports, (ii) reported confounding variables-unadjusted data on the association of peptic ulcer disease with COVID-19 severity were excluded. In the revised manuscript, we have added these. Thank you again for your comments. Discussion

Point 12: In general, you should re-write the Discussion section.

Response 12: Thank you very much for your constructive suggestions and comments. In the revised manuscript, we have rewritten our discussion. Thank you again for your comments.

Round 2

Reviewer 3 Report

Dear Editors,

Since Authors follow Reviewer's instructions, I believe that the manuscript could be published now.